# Geotechnical Properties of Anthropogenic Soils in Road Engineering

**Andrzej Głuchowski** [1,*] , **Katarzyna Gabryś** [1,†] , **Emil Soból** [2,†] , **Raimondas Šadzevičius** [3] **and Wojciech Sas** [1]

1   Water Centre WULS, Warsaw University of Life Sciences-SGGW, 02787 Warsaw, Poland; katarzyna_gabrys@sggw.pl (K.G.); wojciech_sas@sggw.edu.pl (W.S.)
2   Institute of Civil Engineering, Warsaw University of Life Sciences-SGGW, 02787 Warsaw, Poland; emil_sobol@sggw.edu.pl
3   Institute of Hydraulic Engineering, Vytautas Magnus University Agriculture Academy, 53361 Kaunas, Lithuania; raimondas.sadzevicius@vdu.lt
*   Correspondence: andrzej_gluchowski@sggw.edu.pl; Tel.: +48-225-935-405
†   These authors contributed equally to this work.

**Abstract:** The construction of a roads network consumes high amounts of materials. The road materials are required to fulfill high standards like bearing capacity and low settlement susceptibility due to cyclic loading. Therefore, crushed aggregates are the primary subbase construction material. The material-intensity of road engineering leads to depletion of natural resources, and to avoid it, the alternative recycled materials are required to be applied to achieve sustainable development. The anthropogenic soils (AS), which are defined as man-made unbound aggregates, are the response to these requirements. For the successful application of the AS, a series of geotechnical laboratory and field tests were conducted. In this article, we present the set of 58 test results, including California Bearing Ratio (CBR) bearing capacity tests, oedometric tests, and cyclic CBR tests, to characterize the behavior of three AS types and to compare its reaction with natural aggregate (NA). The AS tested in this study are recycled concrete aggregate (RCA), fly ash and bottom ash mix (BS), and blast furnace slag (BFS). The results of the tests show that the AS has similar characteristics to NA, and in some cases, like compression characteristic, RCA and BFS behave a stiffer response to cyclic loading. The test results and analysis presented here extend the knowledge about AS compressibility and AS response to cyclic loading.

**Keywords:** anthropogenic soil; natural aggregate; recycled concrete aggregate; blast furnace slag; bottom slag; CBR; cyclic loading; cCBR; oedometric tests; optimal moisture content; geotechnics

## 1. Introduction

The constant increase of investments in the construction industry leads to grooving production of man-made material (MM) unbound aggregate. This type of material is mainly stored in landfills and highly contributes to gross waste products around the globe. One of the methods which deals with MM materials is recycling as the production of aggregates for use in concrete [1–3]. The second target of MM wastes recycling is application as a base and a subbase in road construction layers [4–7]. This type of soil in road engineering is not only limited to the construction layers but can be used as an embankment fill or as a supplement to soil stabilization [8,9].

The reuse of reclaimed construction materials, as well as post-combustion slag or steel slag, is essential from the sustainability point of view. The use of nonrenewable resources such as natural aggregates (NA) is undesirable, and substitutes are highly recommended, especially in such constructions as roads [10].

The reclaimed concrete and slag materials such as MM soils may differ significantly from NA. Therefore, in the soil classification, they are separated from NA and have been named as anthropogenic soils (AS) [11]. The successful application of AS in road engineering needs to be supported by numerous tests such as the California Bearing Ratio (CBR) and shear strength tests to evaluate the bearing capacity of AS. This effort is essential to establish how far the AS properties differ from the NA properties, despite the origin. As a matter of fact, many efforts were put into this kind of test. For example, CBR value was reported for almost all popular AS in different moisture content conditions and different gradation curve cases [12–20]. For example, steel slag addition to natural soil mix improves the CBR value by 10% [12], and the CBR for the 25% steel slag-clay mix has shown value equal to 90% [13]. The colliery spoils and fly ash mix composite with the 10% share of ash in mix have shown the CBR value equal to 43–48% in soaked and 36–42% in unsoaked conditions [14,15]. The soft soils with a variation of granulated blast furnace slag and fly ash addition have shown an increase of the soil CBR value from 8% for natural soil to 18% for 9% blast furnace slag and 3% fly ash mix [16]. The recycled concrete aggregate reported that the CBR value is usually at the same level or even higher than for natural soils. For example, for the soil with sandy gravel grain composition, the CBR was in the range of 36% to 44% in [17] and in the range of 118% to 160% in [18]. Recycled concrete aggregate is often used in soil mix. The study of recycled concrete aggregate mix with Recycled Asphalt Pavement (RAP) has shown that the 85% RCA –15% RAP mix CBR value is equal to 66%. It enables the application of this mix in road construction and to utilize the recycled asphalt pavement material with low CBR value (33%) [19]. The same procedure was used for the recycled concrete aggregate and Crushed Clay Brick (CCB) mix, where the CBR was equal to 62% for the 75% RCA–25% CCB mix [20]. The CBR test is the most popular laboratory test in road engineering when the evaluation of bearing capacity is performed [21,22]. Nevertheless, the CBR test is continuously criticized for being too simple to evaluate actual soil-bearing capacity characteristics. Despite these objections, the CBR test is often only available in road engineering laboratories, and its simplicity, as well as its repetitiveness, make it an important test to judge whether the aggregates can be used in the desired road layer. Some developments of the CBR techniques were recently conducted to link the test results with classic mechanic parameters of tested material [23].

One of the CBR test modifications is the cyclic CBR (cCBR) or repeating load CBR. The principle of this test was presented by Araya et al. [21,24,25], and the purpose of this test is to obtain the resilient characteristics of unbound granular materials (UGM). In order to do so, the soil is loaded with standard CBR procedure to the depth of plunger penetration equal to 2.54 mm, and the force of loading is then repeated around 50 times to achieve the resilient soil response.

The cCBR test was conducted on UGM, stabilized soils, and AS in which the resilient modulus ($M_r$) was derived. The cCBR has proven to be a reliable method for $M_r$ value evaluation [26–28]. The standard CBR correlations with resilient modulus have been obtained recently and show that the soil modulus value is dependent on the particle shape, the moisture, saturation conditions, and on the compaction energy [29–34].

The CBR test relays on the plunger penetration into elastic half-space. Based on the distribution of pressure under the plunger $\sigma_z(r)$ presented by Sneddon [35], in Equation (1):

$$\sigma_z(r) = \frac{\pi \times a^2}{2F \times \sqrt{1 - \left(\frac{r}{a}\right)^2}} \tag{1}$$

where $F$ is the penetration force during the CBR test, $a$ is the radius of the plunger, and $r$ is the distance from the plunger center, the vertical stress has a singularity at $r = an$ in which vertical strain has an infinite value. This assumption enables us to assume that the plasticity is localized around the plunger edge. The rest of the sample remains in the elastic zone. During static loading, the soils have to be treated as the elasto-plastic medium, which complicates proper analysis of elastic modulus. An ideal condition, in which the investigation would be performed, is a one-dimensional oedometric test (since

there is no horizontal soil movement allowed) or some kind of CBR test in which no plastic effect occurs during loading. Therefore, the cCBR test is the best answer to these problems. During the repeating load, the soil plastic strain decreases to the point where the resilient response is observed. In such conditions, the measured modulus represents the elastic response of the soil sample.

The CBR value and Young's modulus relation were analyzed by Mendoza and Caicedo [33]. The results of CBR tests on granular soils have shown that the correlation between CBR and elastic modulus has somewhat nonlinear characteristics. The CBR decreases when the yield stress decreases with nonlinear characteristics. What is more, the results of the CBR test on different soils show that the CBR value depends on the quality and the origin of the granular material. Therefore, the CBR depends on particle size and shape, as well as on the particle breakage susceptibility. The finite element method simulations conducted in this article show that, during the CBR test, the compression paths have a significant share among all stress paths with a much lesser percentage of shear paths. The yield is mobilized, therefore, due to compression in which the cap yield surface is moved by stress path. The results of calculations show that any relationship between CBR and elastic modulus should be treated with reserve since, for low pressure, the elastic modulus remains constant.

Among rich literature covering the topic of elastic modulus calculation from the CBR test, a few are mentioned below. Nielson et al. [36] derived Equation (2) based on classical soil mechanics relationships:

$$E = \frac{0.75 \times \pi \times a(1-v)^2}{(1-2v)} \times 689.5 \times CBR, \tag{2}$$

where $a$ (plunger area) is in (inches), and $v$ is Poisson's ratio, and $E$ is Young's modulus (kPa). The Guide for Mechanistic-Empirical Design of New and Rehabilitated Pavement Structures gives the following relationship between the CBR and Young modulus, in Equation (3):

$$E = 17.6 \times CBR^2, \tag{3}$$

where $E$ is Young's modulus (MPa).

In this article, three types of anthropogenic soils and one type of natural aggregate were tested to derive the relationship between the cCBR elastic modulus and the oedometric compression parameters. The four soil materials had a standard gradation composition to enable the comparison of the properties between them. In this study, we analyzed if, from the cCBR test, the direct relationship to classical mechanics' parameters and the CBR value can be established for the natural and anthropogenic soils.

## 2. Materials and Methods

### 2.1. Soil Properties and Sample Preparation

In this research, one kind of natural soil, i.e., limestone crushed stone as natural aggregate (NA), and three kinds of anthropogenic soils, i.e., recycled concrete aggregate (RCA), fly ash and bottom ash mix (BS) and blast furnace slag (BFS), were used in laboratory tests (Figure 1). A brief description of each material is presented below. Additionally, a unified granulometric composition was created for each soil. Moreover, the soil fractions for a given soil type were mixed to get the grain size distribution curve common for each soil type.

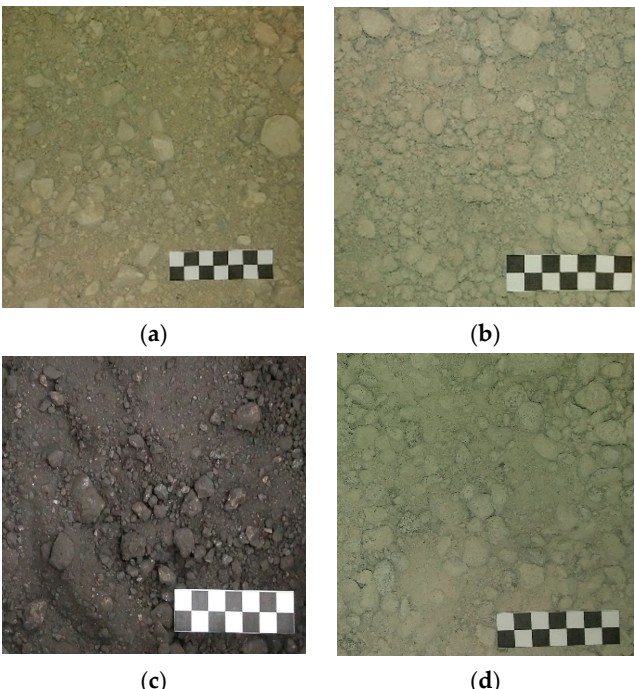

**Figure 1.** Materials tested: (**a**) crushed limestone, (**b**) recycled concrete aggregate, (**c**) fly ash and bottom ash mix, (**d**) blast furnace slag (the side of the scale in the form of a chessboard has a length of 1 cm).

### 2.1.1. Crushed Limestone

Crushed limestone is a form of construction NA, typically produced by mining rock deposit and breaking the removed rock down to the desired size using a crusher. NA is one of the most accessible natural resources. Its advantage is that it can be crushed and sized to meet most specifications. This material is clean, angular, and binds well with cementing mixtures. A uniform lithological composition can be maintained with little or no selective quarrying in many areas. The specific gravity of NA is equal to 2.67.

### 2.1.2. Recycled Concrete Aggregate

As a result of the separation process and later crushing of concrete products, recycled concrete aggregates (RCA) are produced. Despite differences during the tests, factors such as bearing capacity or shear strength have shown an excellent performance of RCA, meaning that it can be applied as a pavement subbase [17]. As an artificial aggregate, RCA characterizes another structure composition in comparison with natural aggregates. RCA contains portions of the cement matrix that consists of anhydrous cement and hydration products, which form a porous microstructure. The high water absorption of the RCA, especially outer layers called 'attached mortar,' may be the reason for lower mechanical properties when compared to natural aggregates [36,37].

RCA in this research was taken from a building demolition site in Warsaw, by the skid-mounted impact crusher. The strength class properties of the construction concrete, made from Portland cement, were estimated at the level from C16/20 to C30/35, based on the data obtained from building plans. Then, the material was fractionated using a mechanical shaker and divided into several types, each one composed of sieved fractions. In this study, only one type with grain diameter dimensions 0–20 mm was investigated in the laboratory. The aggregates were 99% composed from broken cement concrete, the rest being glass and brick ($\Sigma$(Rb, Rg, X) $\leq$ 1% m/m), under standard [38,39], and contained no asphalt or tar elements. The specific gravity of RCA in this study is equal to 2.54.

### 2.1.3. Fly Ash and Bottom Ash Mix

Fly ash and bottom ash (BS) are the solid residue byproducts produced by coal-burning electric utilities. They are mainly disposed of together as waste in utility disposal sites, where a typical disposal rate for fly ash is about 80% and for bottom ash is about 20% [40]. Fly ash is the finely divided residue resulting from the combustion of powdered coal, which is transported from the firebox to the boiler by flue gas, whereas bottom ash is a byproduct of burning coal at the thermal power plant. Bottom ash particles are much coarser than the fly ash. It is a coarse, angular material of porous surface texture, predominantly sand-sized [41]. The chemical composition of both materials is similar, but bottom ash typically contains a greater quantity of carbon. It exhibits as well high shear strength and low compressibility [41].

In Poland, approximately 50 million tons of hard coal and 60 million tons of brown coal are used annually to generate electricity. When burning such an amount of coal, vast amounts of slag, ashes, and fly ash are generated. Each year, about 24 million tons of this waste are produced. The amount of fly ash produced over the year, for example, has changed over the last 9 years, ranging from a maximum of 4.6 million tons in 2012 down to 3.3 million tons in 2016 and 2017 [42]. The combustion waste from energy generation is primarily used in the production of construction materials (including cement), the construction of roads, and mining [42]. BS is a material that finds its application in road construction. The demand for this material increases year by year due to the increase in communication investments, plans to expand the motorway network in Poland, and a change in the design of communication routes in cities that are currently carried out on earth embankments instead of using reinforced concrete flyovers.

The usefulness of BS depends on the following features: structure, degree of sintering, and content of ingredients. Its basic characterization for application in the improved substrate, i.e., reinforcing, frost-resistant, and drainage (filtration) layers, are high internal friction angle, low bulk density, as well as high filtration at every operating time [43].

BS used in this research was supplied by the Siekierki Cogeneration Plant from Warsaw, which is a part of the power station PGNiG TERMIKA. Byproducts of combustion (ash and slag) at the PGNiG TERMIKA plants are regularly transferred to recipients possessing the appropriate licenses. They are used in the following areas: road construction, in the production of construction materials, and as leveling material. The ash and slag generated in large volumes during the peak heating period are stored and then recovered in the summer [42].

The fly ash and bottom ash were first dried for 24 h and brought to room temperature. Next, they were mixed together in the required proportions, and their geotechnical characteristics were investigated. The specific gravity of AF + BA in this study is equal to 2.03. A similar value was obtained by [44].

### 2.1.4. Blast Furnace Slag

Blast furnace slag (BFS) is a byproduct of the steel manufacturing industry [45] with a long tradition of use in different areas of civil engineering. BFS is formed when iron ore or iron pellets, coke, and a flux (either limestone or dolomite) are melted together in a blast furnace. When the metallurgical smelting process is complete, the lime in the flux has been chemically combined with the aluminates and silicates of the ore and coke ash to form a nonmetallic product called blast furnace slag. During the period of cooling and hardening from its molten state, BFS can be cooled in several ways to create any of several types of BF slag products. In most cases, the slag is cooled by water, dried, and ground to a fine powder. This ground granulated blast furnace slag (GGBFS) contains over 95% glass and has high reactivity. When cooling takes place more slowly, the glass fraction decreases, leading to a significant reduction of the reactivity when crystalline minerals are present [46].

Granulated blast furnace slag has been applied in the cement industry for over 100 years. Nowadays, slag is widely used in the cement and concrete industry, and over 16 million tons per year, granulated slag is produced by the European steel industry (website: Orcem). In this study, examined BFS has a hydraulic property, and there is no risk of alkali-aggregate reaction. Because of the potent

latent hydraulic property that results from fine grinding, BFS is applied in products such as Portland cement. When blended with cement, GGBFS becomes Portland blast furnace slag cement (PSC) with the same properties as ordinary (Portland) cement. PSC is a mixture of ordinary Portland cement and not more than 65 wt % of granulated slag. It is generally recognized that the rate of hardening of slag cement is slower than that of ordinary Portland cement during the first 28 days, but after that, increases so that, at 12 months, the strengths become close to or even exceed those of Portland cement [47]. The advantages of this blast furnace slag cement, such as briefly mentioned increasing strength over long periods, low heating speed when reacting with water, and high chemical durability, are put to practical use in a broad range of fields, including the construction of ports and harbors and other significant civil engineering works [48].

BFS in this research was obtained from the Lafarge Cement SA cement plant, created in the blast furnace process. One type gained as a result of sieve analysis, with a grain diameter of 0–20 mm, was used in the tests. The specific gravity of BFS in this study is equal to 2.58.

## 2.2. Static CBR Tests

We performed CBR tests to determine the engineering properties of the anthropogenic soils used in road applications. The CBR test is a penetration test applied to evaluate the subgrade strength of roads and pavements. In this test, a standard piston, with a diameter of 50 mm, penetrated the soil at the standard rate of 1.25 mm/minute. The pressure up to penetration of 2.5 mm was measured, and its ratio to the bearing value of a standard crushed rock was termed as CBR.

Our laboratory CBR tests were conducted to the Proctor method, according to the American Society for Testing and Materials (ASTM)standard [49]. The selected mode is characterized by the use of a 2.5 kg hammer and a larger mold of 150 mm diameter and 120 mm height, with a volume of 2.2 dm$^3$. A 3-layer Proctor test was performed, with 56 blows to each layer. This procedure creates constant energy of compaction, whose level is equal to 0.59 J/cm$^3$. We also did the vibratory compaction tests with the use of a vibratory compaction hammer. The compaction was conducted in three layers, and 8 s excitation on each layer was applied. The energy of compaction corresponds with the Proctors test energy of compaction. The preliminary tests highlighted though the maximum dry density and optimum moisture content (OMC).

## 2.3. Cyclic CBR Tests

Another mechanical property analysis was the cyclic CBR test. The cCBR tests were conducted on samples prepared in CBR mold by applying the cCBR test procedure. The idea behind the cCBR test comes from the CBR test popularity and uncomplicated test procedure.

The principles of this test during the first step were the same as for the usual CBR test. Each specimen was loaded by force of 0.05 kPa in order to keep in contact with the plunger from the beginning of the test. The test started under standard conditions, i.e., penetration velocity of 1.27 mm/min (the frequency equals to 0.0084 Hz), the penetration depth of 2.54 mm in the first cycle. When the desired plunger penetration was reached, the unloading phase was performed to stress equal to 10% of the maximal stress noted on 2.54 mm penetration. The first loading cycles consisted of the loading and unloading phases. The next cycles were performed to the maximal and minimal stress obtained in the first cycle. The number of cycles was determined by the percentage of the plastic displacements in one cycle (usually 50 to 100 cycles). The test can be considered complete when, in one cycle, less than 1% of noted displacements are plastic [50].

Hence, in summary, the cCBR test is aimed to simulate displacements of subgrade surface by constant force application and to observe plastic displacements behavior under cyclic loading [51].

## 2.4. Modified Oedometer Tests

The oedometer consolidation test was adopted in this study for the determination of the compressibility of the tested materials when subjected to vertical loads. The results were used

to calculate and estimate the compression index $C_c$ and preconsolidation pressure $p'_c$. The compression index is the parameter that can be applied to settlement calculations.

In this test, compacted soil specimens were loaded axially in constant stress steps until the primary consolidation ceased. The important aim of modified oedometer tests was to obtain compressibility characteristics of the tested materials in the moisture content in which the samples were compacted. The oedometric tests of our specimens took place in the Proctor cylinder (d = 150 mm, h = 120 mm), which supports no lateral movements of the soil. The following sequence of loading step were applied: 12.5, 25, 50, 100, 200, 400, 800, 1600 kPa. Each increment of loading was held constant for 100 s. This time procedure was chosen to get the compression characteristics and to avoid any additional soil deformation due to time-effect concerning the grain crushing, which is herein called secondary compression. Therefore, in this study, the compression curves only characterized the skeleton compression in first phase of the loading. In Appendix A, we present a raw data example of the displacement-log(t) characteristic, indicating sharp limit between primary and secondary compression observed in all tested samples. After the first loading, the unloading process was done in one step to the pressure of 50 kPa. Next, the reloading process was initiated to the value of 1600 kPa, and finally again first loading up to 3200 kPa.

## 3. Results

### 3.1. Soil Gradation Curve

The four types of soil tested in this article were sieved, and then the standard soil gradation curve 0–20 mm was composed based on the weight share. The gradation curve fulfills the requirements of Polish [52], English [53], and American [54] codes, and the soil with such gradation can be used as a road subbase material. The common gradation curve is presented in Figure 2. The coefficient of curvature ($C_C$) and coefficient of uniformity ($C_U$) were calculated in order to classify the shape of the grading curve of tested types. The $C_U$ value is equal to 30.0, and $C_C$ is equal to 0.53. The soil was, therefore, classified as well-graded, according to Eurocode 7 [55–57]. The fractions composition of the tested gradation curve has led to recognizing the soil as gravelly sand (grSa), according to [56,57], and as well-graded sand or gravelly sand (SW), according to Unified Soil Classification System (USCS) [58].

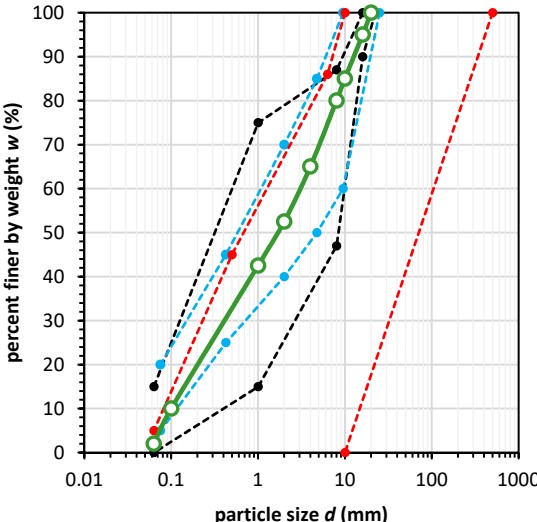

**Figure 2.** Soil gradation curve for Natural Aggregate (NA) and Anthropogenic Soils (AS) tested in this study (solid green line, the tested soil material was first sieved into fractions, and then the universal soil grain composition was created for each kind of soil), the red dashed line-British standards requirement, blue dashed line-American standards requirement, black sashed line Polish standards requirement.

## 3.2. Optimum Moisture Content

The compaction of four soil types consists of the Proctor test and vibratory hammer tests. The test results show high dependence of compaction test type on the test results. The soil tested with the Proctor method had an inconsistent characteristic, and no apparent impact of moisture content can be seen. In Figure 3, the results of the compaction tests are presented. The NA reaches the highest density in moisture content equal to 6.52%, which corresponds to the saturation ratio equal to 0.94, and the maximal dry density is equal to 2120.0 kg/m³. The gravelly sands or sandy gravels usually have a higher dry density in low moisture contents, and with an increase of water content, the density drops. The compaction curve convex is upwards. When the effect of matric suction decreases due to increased moisture, the dry density rises, which we can observe in the case of NA (Figure 3a).

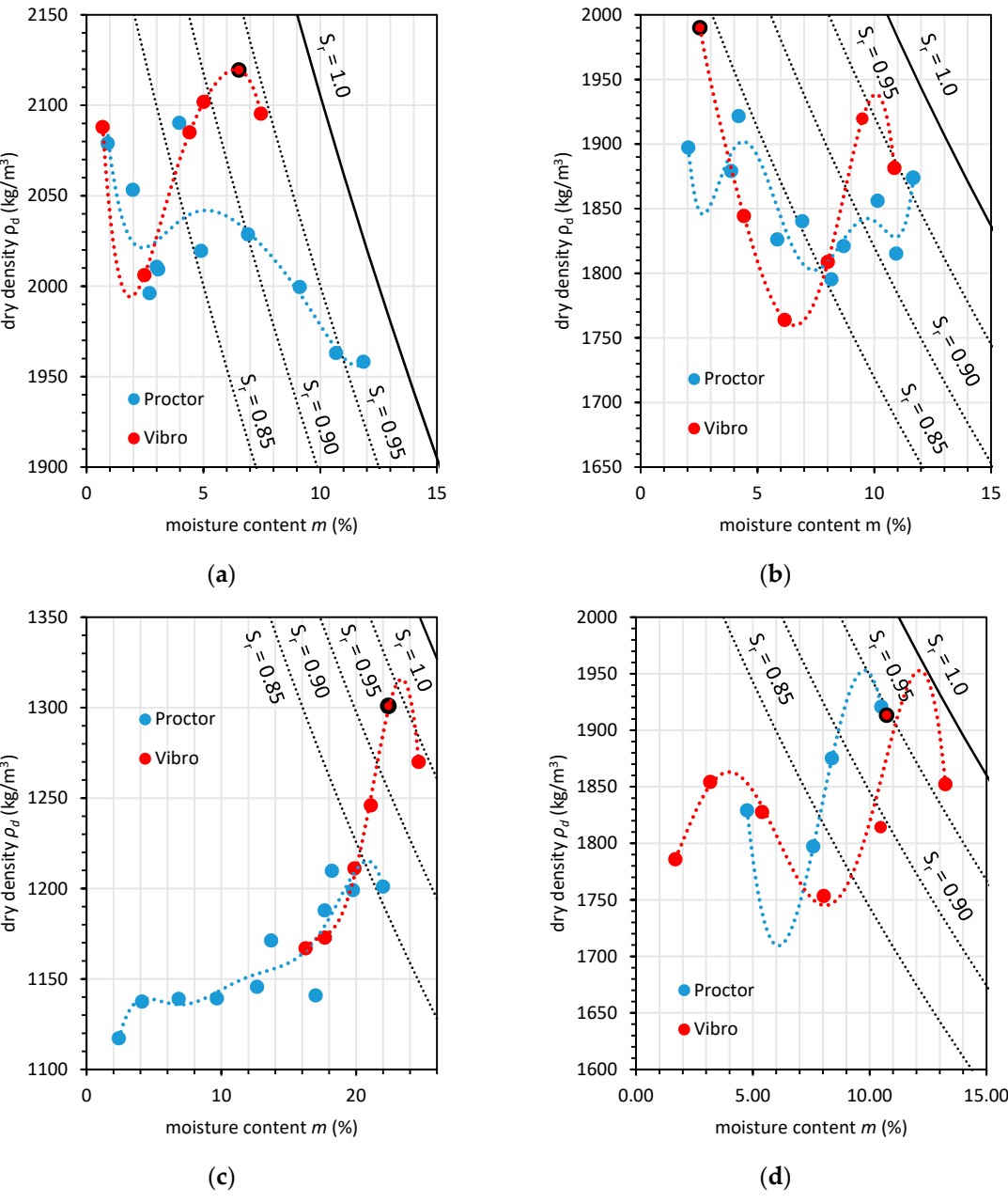

**Figure 3.** Results of compaction tests for (**a**) NA, (**b**) Recycled Concrete Aggregate (RCA), (**c**) fly ash and bottom ash mix (BS), (**d**) blast furnace slag (BFS). Points with black edge indicate the point of maximum dry density and Optimum Moisture Content (dotted lines-compaction curves).

The anthropogenic soils behave quite differently. For example, the RCA has the highest density in the lowest tested moisture content. This conclusion can be drawn based on both vibro-compaction and Proctor compaction tests. The possible reason for that is the grain roughens, which is hard to estimate. Based on the tests performed by He et al. [59] concerning the interface effects between RCA particles, it was concluded that between RCA grains exist very low values of tangential stiffness. The authors stated that RCA grains have low hardness and soft nature, which contributed to the test results. The surface smoothening and debris production impact the surfaces and result in complex effects of preshearing. The grain shearing shows a reduction of surface roughness and damage to micro-asperities.

This behavior indicates that the RCA grain surface can undergo changes that can lead to different compaction characteristics. The moisture content and matric suction impact result in a decrease of dry density, and even in the state of high moisture content, the RCA does not have a higher dry density than in air-dried conditions.

The BS compaction curve shows that this material in opposite to two previous types can reach a high moisture content at which the maximum dry density is observed. The fly ash and bottom ash mix in such conditions have the lowest dry density from all four soil types tested in this article. The reason for that is low specific gravity, which is caused by high internal porosity grains in the "popcorn-like" shapes. What is more, the dry density in air-dried conditions has the lowest value, which differs this type of soil from the rest of the tested ones. The intermediate moisture content characterizes with moderate dry density. BS reaches dry density equal to 1300.9 kg/m$^3$ in optimum moisture content equal to 22.4%. Similar to previous cases, the vibro-compaction gives higher dry density results in comparison to the Proctor method. The moisture content in such conditions indicates that the fly ash and bottom ash mix is in near full saturation state ($S_r \approx 0.94$). Therefore, during the field compaction, BS should be in "flushed" conditions to obtain the highest degree of densification.

The BFS (Figure 3d) compaction has the compaction characteristics close to RCA, but in air-dried conditions, the dry density is significantly lower in comparison to the dry density in OMC. In Figure 3a–d, we added compaction curves for the Proctor and vibro test method. The compaction curves show proper consistency with the vibro test method. The noncohesive soils have lower susceptibility to the water content than the cohesive soils. This property can be observed as an absence of a typical compaction curve characteristic. The Proctor method was proven to be a less-accurate method for OMC estimation due to the high inconsistency of the test results. The inconsistency is the result of the soil properties. Consequently, we were resigned to use compaction curves to estimate the soil OMC.

### 3.3. Static CBR Test Results

We conducted the static CBR tests on samples compacted with Proctor and vibratory methods. The results are presented in Figure 4. The CBR values were calculated for 2.54 plunger penetration. The highest CBR values for tested soils were observed for vibratory compaction in comparison with the Proctor compaction. The CBR for NA reached 162.4% in optimum moisture content. The RCA has CBR equal to 147.7% in optimum moisture content as well. Lower CBR values were observed in the case of BS (maximum CBR = 42.1%) and BFS (maximum CBR = 93.6%); nevertheless, the top CBR values were observed in OMC. It indicates that the anthropogenic soils are obeying the same rule as NA, where the highest bearing capacity is observed in the OMC conditions. The relation between the anthropogenic soil parameters can be evaluated using the Pearson correlation rang, which gives information about the linear relationship between the soil properties and the CBR value. The results of the calculations are presented in Table 1. The results of the calculations indicate that there exists a significant relationship between CBR parameter change and the physical properties (*p*-value less than 0.050). However, the correlation is rather weak since the correlation coefficient is less than 0.541, in case of dry density (the correlation coefficient equal to 1 or −1 means perfect linear between two variables).

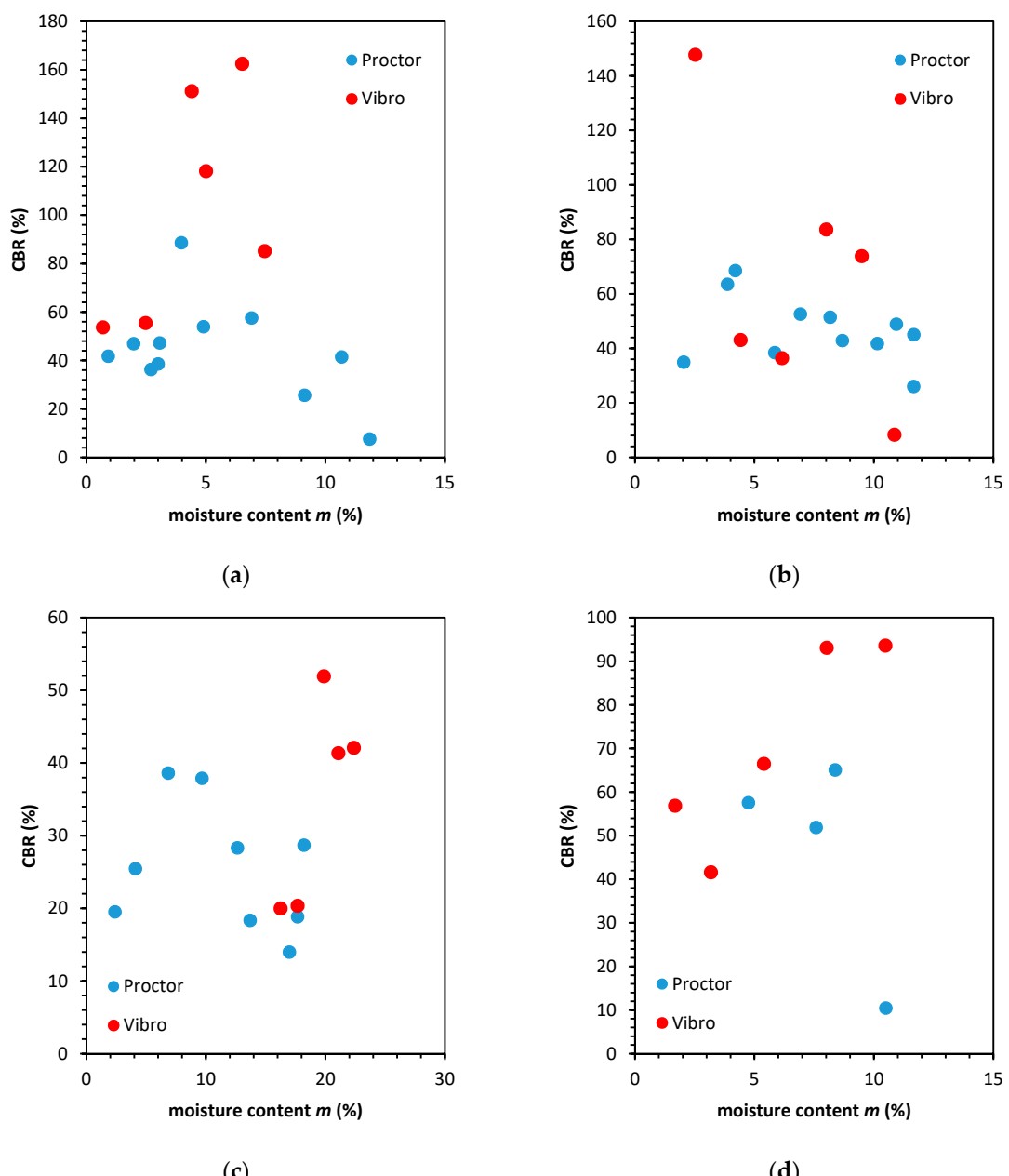

**Figure 4.** CBR value versus moisture content for (**a**) NA, (**b**) RCA, (**c**) BS, (**d**) BFS.

**Table 1.** Pearson correlation analysis result for CBR value.

| | Dry Density $\rho_d$ (kg/m³) | Specific Gravity $S_G$ (–) | Compaction Energy $E_C$ (J/cm³) | California Bearing Ratio (CBR) (%) |
|---|---|---|---|---|
| **Moisture *m* (%)** | −0.565<br>0.00033 | −0.617<br>0.00006 | 0.0997<br>0.563 | −0.357<br>0.0323 |
| **Dry Density $\rho_d$ (kg/m³)** | | 0.987<br>0.0000 | 0.162<br>0.344 | 0.541<br>0.0006 |
| **Specific Gravity $S_G$ (–)** | | | 0.162<br>0.346 | 0.518<br>0.0012 |
| **Compaction Energy $E_C$ (J/cm³)** | | | | 0.373<br>0.02504 |

The underlined number stated for linear correlation as the coefficient of determination $R^2$, the not underlined numbers are the *p*–values where for the *P* greater than 0.050, there is no significant relationship between two variables.

## 3.4. Oedometric Test Results

Most of the geotechnical laboratories are equipped with the Casagrande-type oedometers. This type of oedometer test is suitable for fine graded soils. When the coarse-grained soils are

tested, the larger apparatus and mold are required. This type of apparatus was employed previously. For example, the oedometer apparatus developed by Rowe and Barden is available for specimens with the diameters ranging from 75 mm to 254 mm (BSI 1990) [60,61]. The oedometric tests in this study were conducted in the Proctor cylinder (d = 150 mm, h = 120 mm). The oedometric test results were analyzed, and we have plotted the compressibility curves for all four soil types. For the sake of simplicity, in Figure 5, we plotted the exemplary test result, and Tables 2 and 3 give the results of the oedometer test constants calculation. Some details concerning the compression curves are presented in Appendix B.

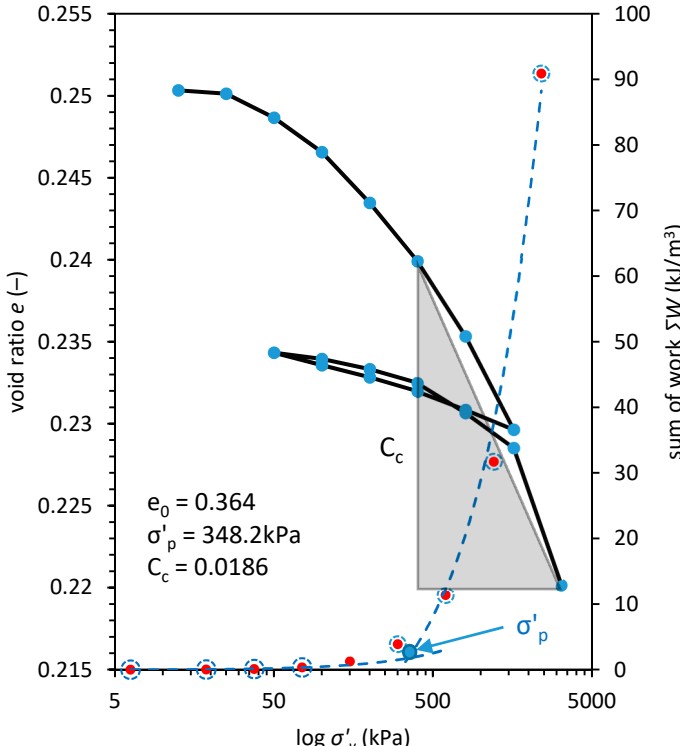

**Figure 5.** Oedometric tests result-compression curve (red points-energy method for preconsolidation pressure calculation). Sample NA *m* = 0.91%.

**Table 2.** Apparent preconsolidation pressure values calculated based on the worn method for four different soil types.

| Natural Aggregate (NA) | | Recycled Concrete Aggregate (RCA) | | Fly Ash and Bottom Ash (BS) | | Blast Furnace Slag (BFS) | |
|---|---|---|---|---|---|---|---|
| Moisture (%) | $\sigma'_p$ (kPa) | Moisture (%) | $\sigma'_p$ (kPa) | Moisture (%) | $\sigma'_p$ (kPa) | Moisture (%) | $\sigma'_p$ (kPa) |
| **Proctor Compaction** | | | | | | | |
| 0.91 | 305.35 | 2.04 | 310.33 | 2.39 | 341.46 | 4.75 | 302.73 |
| 1.98 | 324.23 | 3.87 | 293.26 | 4.11 | 310.76 | 7.58 | 277.87 |
| 2.70 | 341.04 | 4.20 | 310.33 | 6.84 | 346.43 | 8.38 | 315.59 |
| 3.00 | 276.91 | 5.85 | 277.82 | 9.67 | 312.49 | 10.50 | 297.15 |
| 3.07 | 297.33 | 6.92 | 309.47 | 12.64 | 281.09 | | |
| 3.97 | 346.43 | 8.17 | 312.57 | 13.69 | 315.08 | | |
| 4.90 | 337.24 | 8.68 | 311.06 | 16.99 | 326.34 | | |
| 6.91 | 281.09 | 10.14 | 276.80 | 17.67 | 324.23 | | |
| 9.13 | 348.20 | 10.94 | 312.02 | 18.20 | 341.04 | | |
| 10.68 | 310.76 | 11.66 | 292.00 | | | | |
| 11.85 | 326.45 | 11.67 | 310.33 | | | | |
| **Vibratory Compaction** | | | | | | | |
| 0.69 | 367.34 | 2.53 | 342.14 | 16.26 | 367.74 | 1.68 | 500.62 |
| 2.48 | 329.53 | 4.42 | 294.84 | 17.67 | 367.81 | 3.18 | 320.03 |
| 4.41 | 401.26 | 6.16 | 315.08 | 19.88 | 335.82 | 5.40 | 360.61 |
| 5.01 | 262.80 | 8.01 | 245.11 | 21.09 | 347.98 | 8.03 | 340.89 |
| 6.52 | 315.99 | 9.49 | 302.72 | 22.39 | 285.57 | 10.48 | 315.99 |
| 7.46 | 330.21 | 10.86 | 294.84 | | | 10.72 | 310.76 |

**Table 3.** Compression index values for tested soil, calculated based on the result of the oedometric test.

| NA | | RCA | | BS | | BFS | |
|---|---|---|---|---|---|---|---|
| Moisture (%) | $C_C$ (-) | Moisture (%) | $C_C$ (-) | Moisture (%) | $C_C$ (-) | Moisture (%) | $C_C$ (-) |
| **Proctor Compaction** | | | | | | | |
| 0.91 | 0.00351 | 2.04 | 0.00424 | 2.39 | 0.00862 | 4.75 | 0.00397 |
| 1.98 | 0.00372 | 3.87 | 0.00393 | 4.11 | 0.00491 | 7.58 | 0.00415 |
| 2.70 | 0.00395 | 4.20 | 0.00419 | 6.84 | 0.00501 | 8.38 | 0.00391 |
| 3.00 | 0.00395 | 5.85 | 0.00419 | 9.67 | 0.00495 | 10.50 | 0.00394 |
| 3.07 | 0.00390 | 6.92 | 0.00484 | 12.64 | 0.00505 | | |
| 3.97 | 0.00374 | 8.17 | 0.00446 | 13.69 | 0.00501 | | |
| 4.90 | 0.00393 | 8.68 | 0.00443 | 16.99 | 0.00777 | | |
| 6.91 | 0.00387 | 10.14 | 0.00445 | 17.67 | 0.00489 | | |
| 9.13 | 0.00386 | 11.66 | 0.00411 | 18.20 | 0.00474 | | |
| 10.68 | 0.00398 | 11.67 | 0.00441 | | | | |
| 11.85 | 0.00405 | | | | | | |
| **Vibratory Compaction** | | | | | | | |
| 0.69 | 0.00430 | 2.53 | 0.003818 | 16.26 | 0.00788 | 1.68 | 0.00652 |
| 2.48 | 0.00387 | 4.42 | 0.004288 | 17.67 | 0.00782 | 3.18 | 0.00423 |
| 4.41 | 0.00364 | 6.16 | 0.004468 | 19.88 | 0.00761 | 5.40 | 0.00411 |
| 5.01 | 0.00359 | 8.01 | 0.004213 | 21.09 | 0.00787 | 8.03 | 0.00420 |
| 6.52 | 0.00383 | 9.49 | 0.004067 | 22.39 | 0.00746 | 10.48 | 0.00400 |
| 7.46 | 0.00399 | 10.86 | 0.004192 | | | 10.72 | 0.00400 |

Based on the compression characteristics, we estimated the *apparent preconsolidation* pressure $\sigma'_p$.

Performed oedometric tests have shown that the preconsolidation pressure can be calculated. The apparent preconsolidation pressure reported here is caused not by the maximum effective vertical overburden stress but by energy of compaction; therefore, the *apparent* term is appended. The effective stress is as well the energy imposed on the sample and the energy of compaction may have the same effect as a static stress imposed on the sample. The preconsolidation phenomena which is a certain skeleton arrangement able to sustain the energy of maximal loading is different from the vibro-compaction actions. The exact mechanics of vibro-compaction were presented in the Basic of Foundation Design by Fallenius [62] and in an article of Duan et al. [63]. According to the literature, the preconsolidation effect is caused by high horizontal stress in the soil, which are an effect of

strong horizontal pulses. Therefore, we decided to change the name of this phenomena to "apparent preconsolidation" term.

The highest apparent preconsolidation pressure was observed for the soil specimens compacted on the dry site. In this study, to calculate the apparent preconsolidation pressure, we utilized the work method. The reason for that is the gentle curved $e$–$\sigma'$ characteristics in which the graphic methods, like the Casagrande method, are not appropriate. The method of work proposed by Becker et al. [64] is based on the work input in the test. In this method, the sum of work ($\sum W$) is a sum of finite work $\Delta W_i$ in the *i*-th load step during the oedometric test. The $\Delta W_i$ is the product of average stress $\sigma'_{avg}$ on the sample ($\sigma'_{avg} = (\sigma'_i + \sigma'_{i+1})/2$) and strain difference $\Delta\varepsilon$ ($\Delta\varepsilon = (\varepsilon_{i+1} - \varepsilon_i)$).

The apparent preconsolidation pressure $\sigma'_c$ differs when different moisture content test results are compared. We noticed that the highest apparent preconsolidation pressure in almost all cases would be found in air-dry conditions ($\sigma'_p$ over 300 kPa). The apparent preconsolidation pressure characteristics are similar to the CBR versus moisture content characteristics, and this indicates that the moisture content impacts both the apparent preconsolidation conditions and CBR bearing capacity. Nevertheless, this relationship is weak in terms of mathematical modeling (correlation coefficient $R^2 = 0.404$). The intermediate moisture content conditions show that the apparent preconsolidation pressure has its average value around 310 kPa, independently from the compaction technique. The apparent preconsolidation pressure–moisture content characteristics are presented in Figure 6.

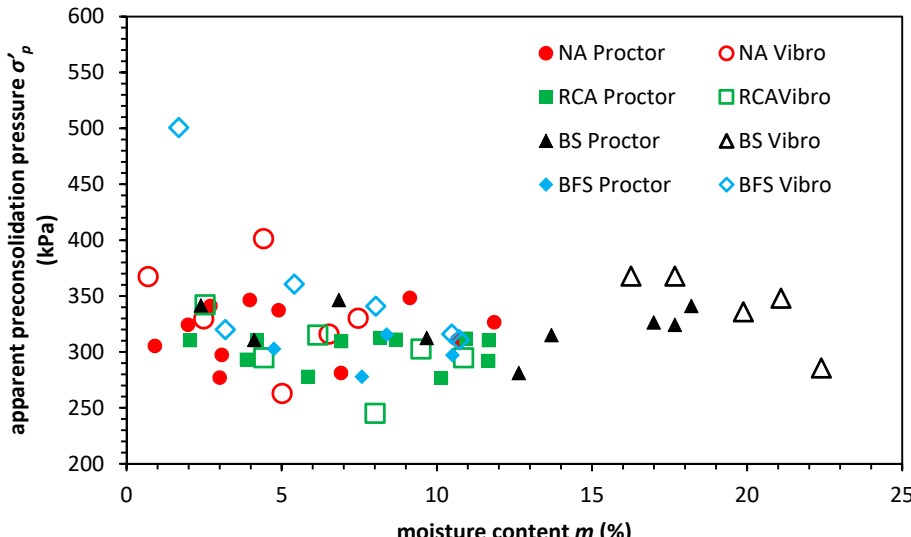

**Figure 6.** Apparent preconsolidation pressure related to the moisture content for anthropogenic and natural soils with different compaction methods, Proctor compaction-full points, vibro-compaction-empty points.

To facilitate the results of compression tests, we calculated the compression index ($C_c$), which is defined as the slope of the virgin consolidation curve. The compression index formula is presented in Equation (4):

$$C_c = \frac{e_0 - e_1}{log\left(\frac{\sigma'_1}{\sigma'_0}\right)} \tag{4}$$

The compression index represents negative gradients on the $e$-log $\sigma'$ plot, but the value is always given as positive. The results of the calculations are presented in Table 3. The value of the compression index is close to the values reported for the dense sand [65].

One of the well-known formulas for $C_c$ calculation, which are based on empirical data, was given by Rendon-Herrero [40] and Azzouz et al. [41], defined as Equations (5) and (6):

$$C_c = 0.29(e_0 - 0.27), \tag{5}$$

$$C_c = 0.43(e_0 - 0.25). \tag{6}$$

Based on the abovementioned relationships, there exists a link between the compression index and the initial void ratio. What is more, the simple linear relationship with the specific gravity ($G_S$) and plasticity index ($I_P$) was also reported [66,67] and is presented as Equation (7):

$$C_c = 0.5G_SI_P \tag{7}$$

Equations (5)–(7) refer to the cohesive soils, but the relationship between the soil physical parameters also applies to noncohesive soil. In Figure 7a, we show the relationship between the compression index and the initial void ratio. The relationship is easy to find. The coefficient of determination $R^2$ for linear regression is equal to 0.415, which means a rather weak fit of linear regression. We performed the multiple linear regression analysis for the pair of independent variables, namely, initial void ratio ($e_0$) and specific gravity ($G_S$), which are proven to have a high correlation with the $C_c$ value (see Equations (5)–(7)). The results of the analysis are presented in Equation (8):

$$C_c = -0.112 + 0.0936e_0 + 0.0392G_S, \tag{8}$$

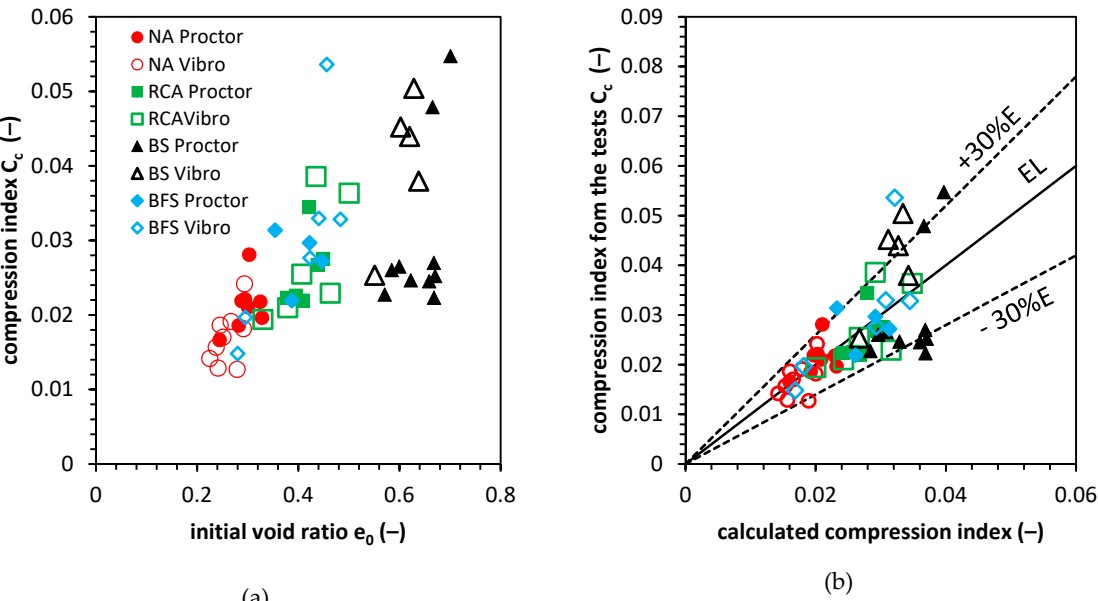

(a)

(b)

**Figure 7.** Compression index relationship to (**a**) initial void ratio, (**b**) results of the calculation with Equation (8) versus test results.

The coefficient of determination $R^2$ is equal to 0.477. The reason for the low fit to the model is the flay ash and bottom ash mix characteristics, which is slightly different from the rest of the tested soils. This relationship is presented in Figure 7b. What is more, with an increase in the initial void ratio, the compression index has a greater range of values, which means that some other factors impact the soil compression characteristics. Therefore, we decided to present additional conditions to Equation (7), which is the result of the data analysis presented in Figure 7b. Most of the results are in the 30% error zone, and therefore, the condition is that the calculated $C_C$ value has to be in a range of ±30%.

Conducted oedometer tests relay on the same sample preparation and molding conditions as the CBR test. Therefore, we did a correlation analysis between the CBR and oedometric test result in terms of the compression index. The results of the calculations are presented in Figure 8.

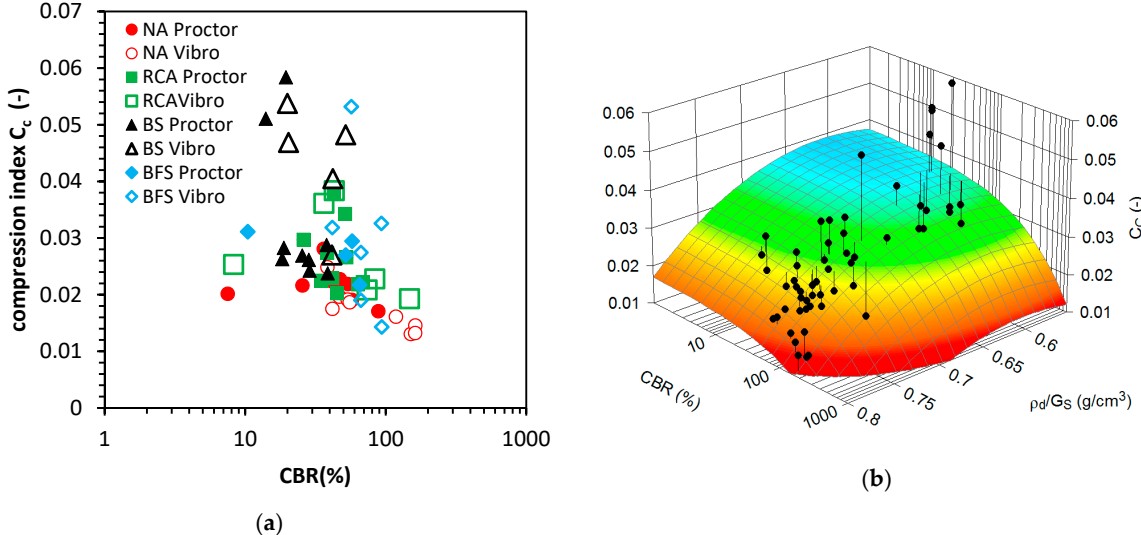

**Figure 8.** Compression index relationship to (**a**) CBR value, (**b**) CBR, and the quotient of dry density and specific gravity.

Based on the Figure 7a analysis, one can find the difference between the NA and the anthropogenic soil characteristics. The NA seems to have more of a strong relationship between CBR and $C_C$ values. The general relationship shows that, with the higher CBR value, the smaller compression index occurs, which is true since the compression index describes the compression characteristics during the CBR test as well. This relationship weakens with the decrease of the CBR value. For the soils with a higher compressibility index, like for RCA or NA, the bearing capacity is also presumed to be higher, and therefore, the CBR value may reflect more compressibility relationship rather than the bearing capacity characteristics.

In order to model the compression index based on the CBR value, we introduced one more factor which impacts the $C_C$–CBR relationship, which is the soil density in reference to the specific gravity. The idea was to check how far the actual soil dry density is from the density of the soil skeleton. The results of the analysis are presented in Figure 8b. The data show that the quotient of dry density and specific gravity ($\rho_d/G_S$) arranges the data point into a path that indicates that, with the increase of $\rho_d/G_S$, the compression index, as well as the CBR value, increase. Based on this relationship, we conducted a mathematical regression analysis to find the mathematical model for the $C_c$ (*CBR*, $\rho_d/G_S$) relationship. The results of the calculations are presented in Equation (9), and the 3D surface, which presents the model, is in Figure 7b:

$$C_c = -0.0595 - 0.535\left(\frac{\rho_d}{G_S}\right)^2 \cdot \ln\left(\frac{\rho_d}{G_S}\right) - 0.000115 \cdot \sqrt{CBR} \cdot \ln(CBR) \tag{9}$$

The coefficient of determination $R^2$ is equal to 0.453. For the presented model, the mean error (ME) is equal to $-4.42 \times 10^{-9}$, and the mean percentage error (MPE) is equal to $-5.63\%$. The mean absolute error (MAE) is equal to 0.0057, which indicates that the calculated compression index value with Equation (9) by average will be in the range of MAE. The more useful parameter is the mean absolute percentage error (MAPE), which informs about the percentage error of the model and, for Equation (9), is equal to 20.22%.

### 3.5. Cyclic CBR Test Results

The cyclic CBR tests were conducted based on the CBR tests performed before the cCBR. During the CBR test, the maximal force of plunger penetration was later repeated by 50 to 100 cycles in order to

find such a state in which the plastic strains during one cycle of loading are close to zero, and when the full elastic response would be achieved.

The closed hysteresis loop obtained by such a procedure was further utilized to calculate the resilient modulus $M_r$ value. The $M_r$ value for cCBR test can be calculated based on Equation (10), proposed by Araya [24]:

$$M_r = \frac{1.513 \cdot \left(1 - v^{1.104}\right) \Delta \sigma_p \cdot r}{\Delta u^{1.012}}, \tag{10}$$

where, $v$—Poisson's ratio (—), $\Delta \sigma_p$ is the stress of plunger penetration between minimal and maximal stress in load cycle (MPa), $r$ is the radius of the plunger (mm), $\Delta u$ is the recoverable displacement in one load cycle. The Poisson ratio in this study was set to NA, RCA, BS, and BFS as equal to 0.30, 0.28, 0.25, and 0.33, respectively. In Figure 9, the relation between the CBR and $M_r$ is presented. The $M_r$ highest value of resilient modulus was observed for NA. The anthropogenic soils had higher resilient modulus when compacted with vibratory methods. The relationship presented in Figure 8 shows that the NA, RCA, and BFS have the same dependency on the CBR value. Therefore, the universal equation can be easily derived to calculate the resilient modulus value. Equation (11) presents the relationship between $M_r$ and CBR in the same manner as Equation (3):

$$M_r = 47.86 \cdot CBR^{0.5411} \tag{11}$$

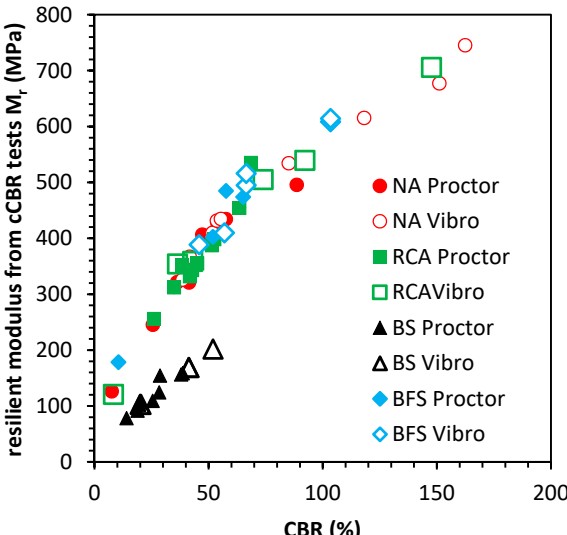

**Figure 9.** Resilient modulus versus CBR value for anthropogenic and natural soil compacted with the use of two methods (Proctor and vibro-compaction).

For this equation, the coefficient of determination $R^2$ is equal to 0.967. The relation between the CBR and resilient modulus value for BS is presented in Equation (12):

$$M_r = 12.58 \cdot CBR^{0.6999} \tag{12}$$

For this equation, the coefficient of determination $R^2$ is equal to 0.955.

If we compare Equations (3), (11), and (12), we would find that the growth of $M_r$ value decreases with the CBR value increase. In the case of Equation (3), we would see the opposite characteristics.

Another relevant characteristic that can be learned from the cCBR test is the increment of plastic displacement characteristics. The soil loaded by stress, which caused 2.54 mm plunger penetration, allows studying what happens when this stress is repeated numerous times. In Figure 10, the

displacement number of cycle characteristics is presented. The characteristics refer to soil displacement development in optimum moisture content.

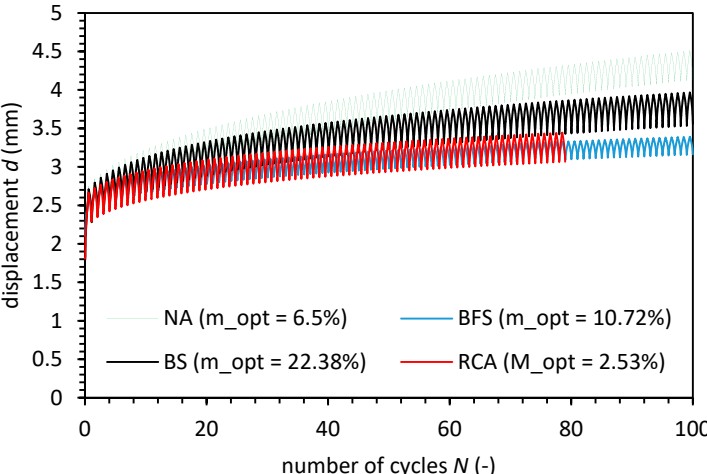

**Figure 10.** Plastic strain development during cyclic CBR test for soils in optimum moisture content.

The cyclic loading takes place a lot longer than the static loading in the oedometer test. The principle of conducting an oedometer compression test is to find the primary compression characteristic. After the primary compression, secondary compression occurs. This kind of compression is occurring in the logarithmic rate, so the slope of the deformation is against the log of time. The secondary compression index $C_{\alpha\_cCBR}$ is computed as Equation (13):

$$C_{\alpha\_cCBR} = \frac{\Delta\varepsilon}{log\left(\frac{N_{100}}{N_{10}}\right)} = \frac{(d_{N=100} - d_{N=10})/h_s}{log\left(\frac{N_{100}}{N_{10}}\right)} \tag{13}$$

We compared the secondary compression index from cCBR tests with the compression index $C_c$ from oedometric tests, and the results of calculations are presented in Figure 11. From the $C_c$ to $C_{\alpha\_cCBR}$ dependence, we might conclude that the vibro-compaction gives higher primary and secondary settlement (both $C_c$ to $C_{\alpha\_cCBR}$ have higher values for this type of compaction). Based on this analysis, the NA has higher compression index parameters than the anthropogenic soils, which means that the anthropogenic soils settle less when loaded with repeating long-term loads. The fly ash and bottom ash mix have slightly different characteristics, and there is no distinct difference between the compaction method.

The overall dependence between $C_c$ and $C_\alpha$ enables to establish the relationship between the soil compressibility index and secondary compression index from the cCBR tests and have the following model (14):

$$C_\alpha = \left[\kappa_1 \cdot e^{\left(-\frac{C_c}{\kappa_2 + \kappa_2 e_0}\right)} + \kappa_3 \cdot e^{\left(-\frac{C_c}{\kappa_4 + \kappa_4 e_0}\right)}\right] \cdot (1 + e_0). \tag{14}$$

The $\kappa_1$ to $\kappa_4$ are constants equal to $\kappa_1 = 0.015545978$, $\kappa_2 = 0.019471099$, $\kappa_3 = 9049380.0$, $\kappa_4 = 0.000512439$ and $e$ is the Euler number. For Equation (14), the coefficient of determination $R^2$ is equal to 0.700. For the model in Equation (14), the mean error (ME) is equal to $-3.85 \cdot 10^{-5}$, and the mean percentage error (MPE) is equal to −21.0%. The mean absolute error (MAE) is equal to 0.0027, which indicates that the calculated $C_\alpha$ from cCBR test value by average will be in the range of MAE. The mean absolute percentage error (MAPE) is equal to 46.18%.

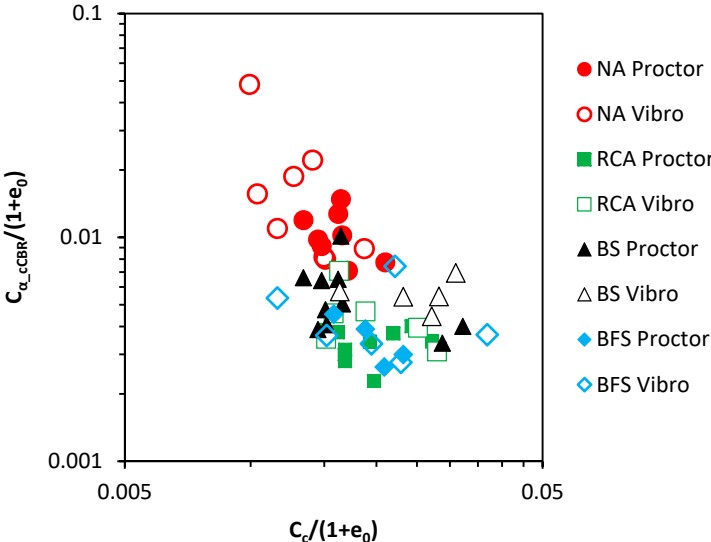

**Figure 11.** The normalized secondary compression coefficient from cCBR tests $C_{\alpha-cCBR}$ versus normalized compression index $C_c$ from the oedometer test.

With the information about the soil secondary compression index for cyclic loading, one can calculate the field secondary settlement $\Delta H_s$ based on the following Equation (15):

$$\Delta H_s = H_f \cdot C_\alpha \cdot log\left(\frac{N_{end}}{N_0}\right) \tag{15}$$

where $N_0$ and $N_{end}$ are the number of cycles at the beginning and end of settlement analysis, and $H_f$ is the thickness of the analyzed soil layer.

## 4. Conclusions

In this article, a set of tests was conducted to characterize the geotechnical parameters of anthropogenic soil and to compare the calculated properties with the geotechnical properties of natural aggregate. The anthropogenic soil in this article is called the soil deposits, whose origin is not natural, but the soil material is man-made. The tested soils were recycled concrete aggregate, fly ash and bottom ash, and blast furnace slag. Based on the test results and the data analysis, the following conclusions, for tested soils, can be drawn:

1. In this article, two methods of compaction were used: the impact compaction, which was conducted to the Proctor method, and the vibratory hammer compaction. The results clearly show that, for the vibratory compaction method, the compaction curve characteristics are more consistent, and the optimal moisture content can be found much easier than in the case of the Proctor method. The compaction curve shows that the highest dry density is achieved for the soil in the air-dry state (for RCA, in this state, the obtained dry density was the highest) and for nearly saturated soil (saturation ratio $S_r \approx 0.95$). Therefore, we can recommend conducting compaction procedure on the anthropogenic soil in a wet state, which is more favorable since, in intermediate moisture content, the dry density in all cases was the lowest. The recommended compaction method is vibratory-compaction as well in all four tested soil cases. The higher dry density might be caused by higher energy of compaction imposed on the soil sample, despite the effort to be consistent with the Proctor method. Nevertheless, vibratory compaction has an additional advantage, which is grain breakage prevention. The impact method causes moderate (RCA and BFS)-to-high particle breakage, like in the case of BS. That can cause significant changes in the resulting gradation curve.

2. The conducted static CBR tests have shown that, in the OMC, sandy gravel materials present exceptional CBR values. In the case of RCA and NA, the CBR value was over 100% (148% and 162%, respectively). The BFS reached 93%, which is also satisfactory for the subbase bearing capacity requirements. In the case of BS, the highest CBR value was 52%, which is not enough to use this material as a subbase, but as the subgrade. What is important to note is that the presented benefits are for samples compacted with vibratory methods. For the Proctor compacted samples, the CBR values are significantly lower. The CBR tests in this study were conducted in unsoaked conditions that have to be taken into account since, in soaked conditions, the CBR values are lower.

3. The oedometric tests have revealed that the compaction effort creates the *apparent preconsolidation* pressure, which, on average, is between 270 and 370 kPa, and this characteristic is the same for all four soil types. We noted slightly higher values of $\sigma'_p$ for the fly ash and bottom ash mix and, in general, higher values of apparent preconsolidation pressure for vibro-compaction and air-dry moisture content.

4. The compression tests have led to compression index $C_c$ estimation for all four soil types. The value of $C_c$ is in the range of the dense sands value reported in the literature. We observed a high correlation between the initial void ratio $e_0$ and specific gravity $S_G$. Therefore, two mathematical expressions were derived which were able to calculate the compression index based on $e_0$, $S_G$, and CBR value. We noted that a stronger correlation between initial void ratio as well as CBR value and compression index exists when the soil is denser and has higher bearing capacity. The reason for this is that more elastic soil responds to static loading when the void ratio is low, and there is no place for particle rearrangement. For example, for BS where initial void ratio was between 0.55 and 0.70, the $C_c$ value had the greatest inconsistency. Conversely, for NA where $e_0$ was equal to 0.24 to 0.32, we observed the highest consistency of $C_c$ value.

5. The cyclic CBR tests led us to calculate resilient modulus value for tested soil types. The highest $M_r$ value corresponded to the highest CBR value and, therefore, to OMC. For NA, $M_r$ was equal to 745 MPa, and for RCA, 705 MPa, for BS, 201 MPa, and BFS, 609 MPa. We compared $M_r$ to CBR values, and we found a strong correlation between these two values. Based on that, we presented two equations for $M_r$ calculation based on CBR value, which is dedicated to NA, RCA, and BS and second for BS. Both equations refer to well-known power functions form the Mechanistic-Empirical Pavement Design Guide, but the constants in this function and for the functions presented by us are different.

6. The plastic displacement observed during the cCBR test has logarithmic characteristics. This phenomenon was accounted for by us as a secondary compression, which is time-dependent. We calculated the secondary compression index from cCBR tests, and we derived the formula for $C_{\alpha\_cCBR}$ calculation based on the compression index value. The $C_{\alpha\_cCBR}$ value can be later used for settlement calculation of a road layer. The plastic displacement accumulation rate was the greatest for NA, which means that the anthropogenic soils may be less susceptible to the rutting.

**Author Contributions:** Conceptualization, A.G.; methodology, A.G., W.S.; investigation, A.G., E.S., K.G.; writing—original draft preparation, A.G., K.G.; writing—review and editing, E.S., R.Š.; supervision, W.S.; All authors have read and agreed to the published version of the manuscript.

**Funding:** This research received no external funding.

**Conflicts of Interest:** The authors declare no conflict of interest.

## Appendix A

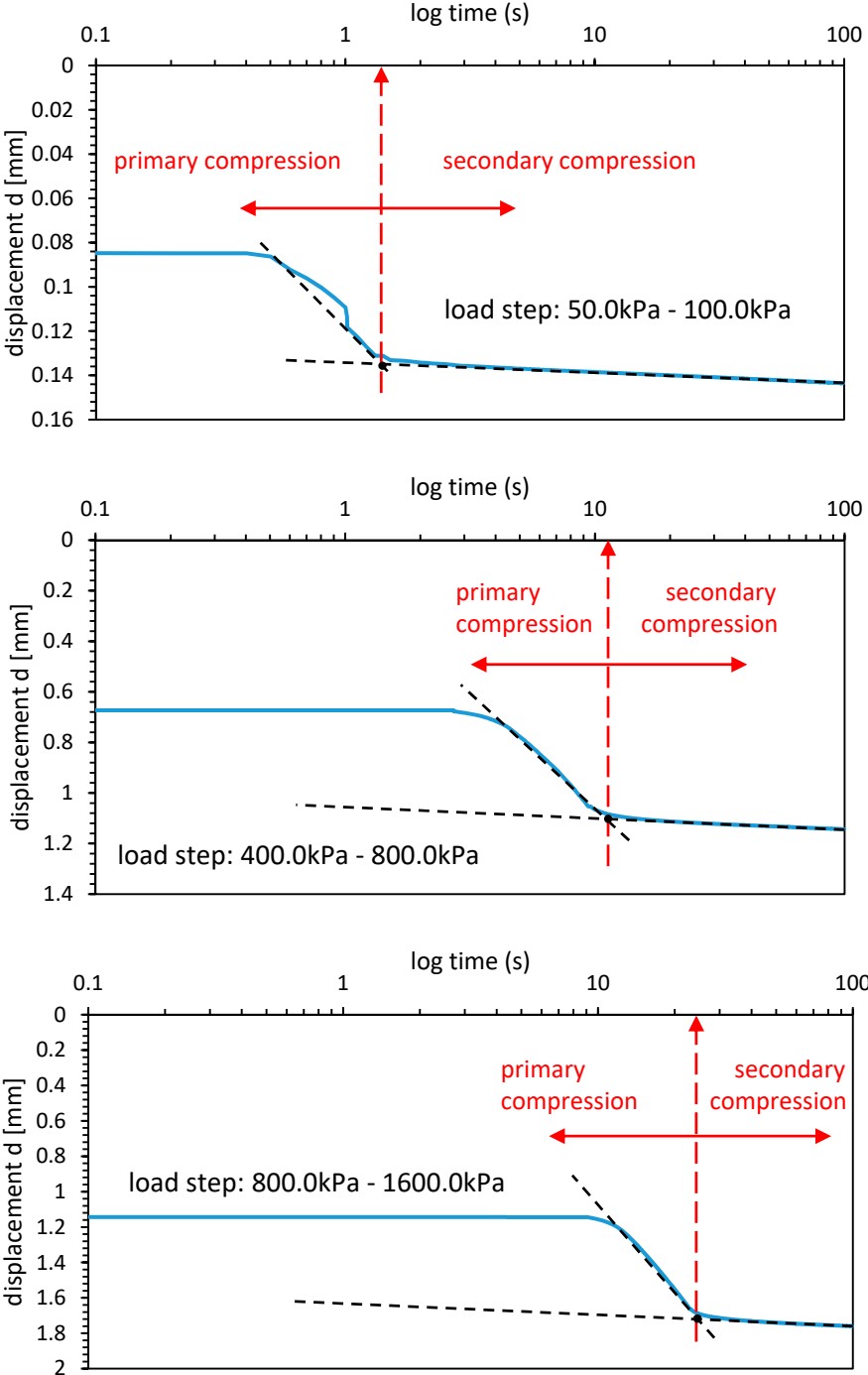

**Figure A1.** Raw data from the oedometer compression test. The figure presents the divide principle between primary and secondary compression in this article.

## Appendix B

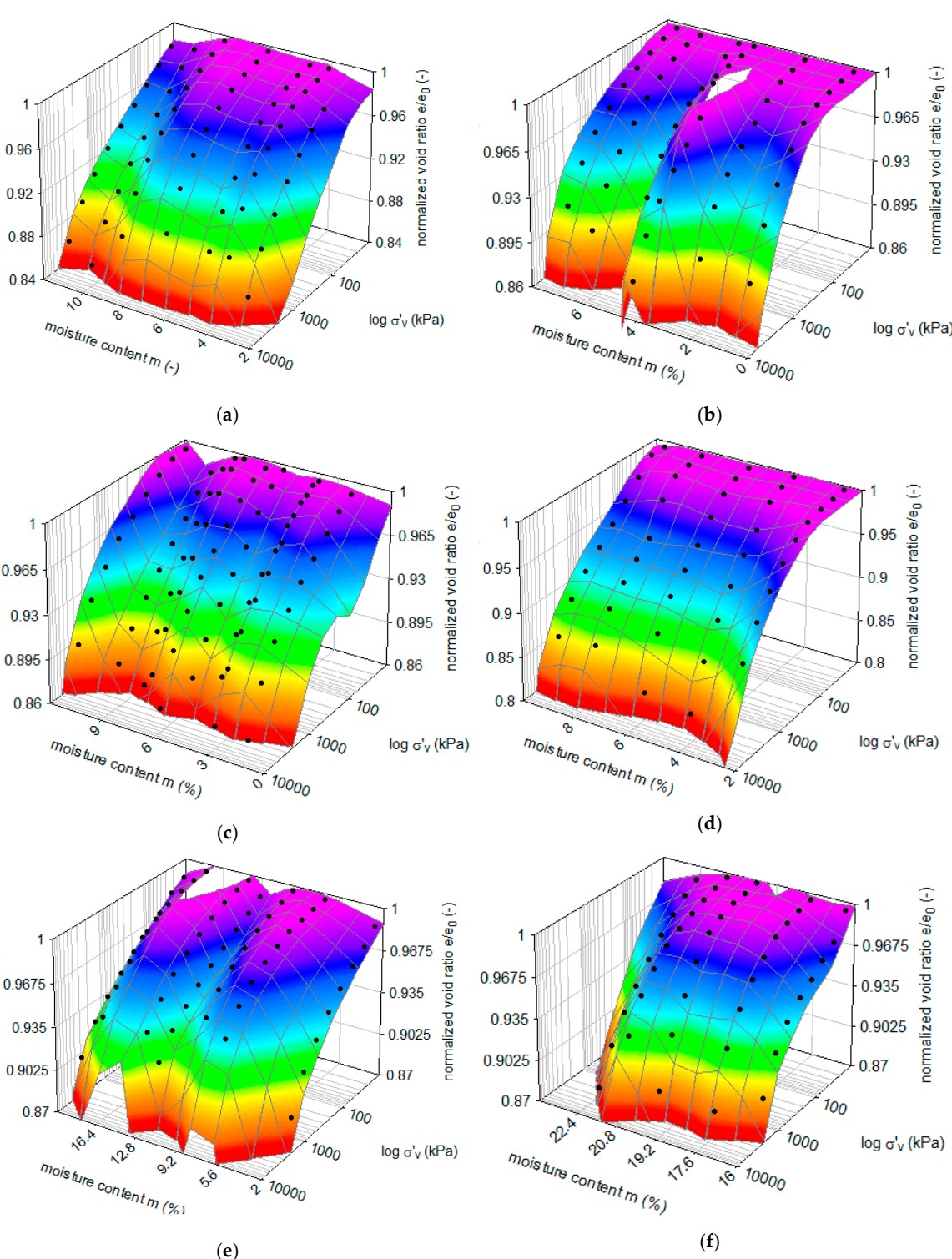

(**a**)

(**b**)

(**c**)

(**d**)

(**e**)

(**f**)

**Figure A2.** *Cont.*

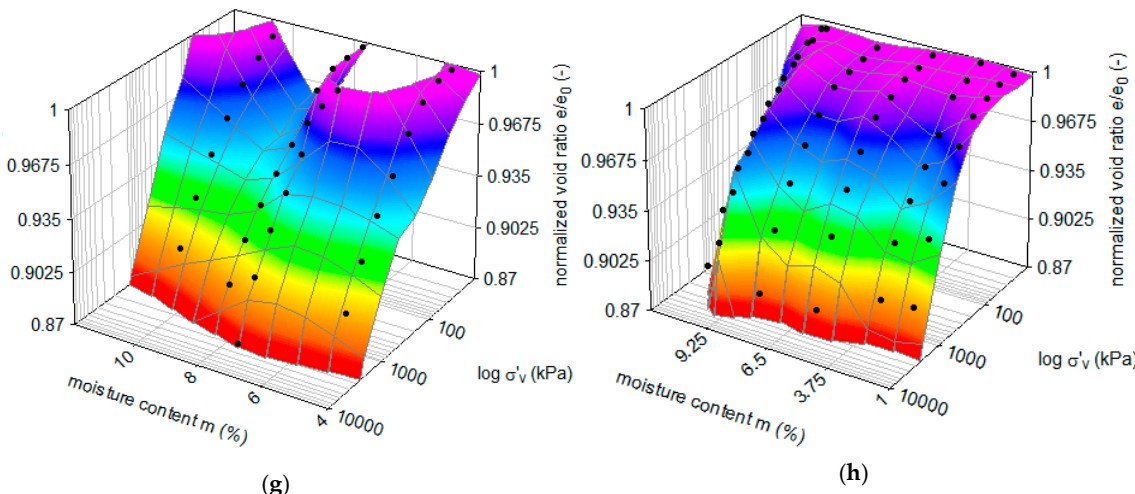

**Figure A2.** Compression curves of soils in different moisture content: (**a**) NA – Proctor compaction, (**b**) NA- vibro-compaction, (**c**) RCA- Proctor compaction, (**d**) RCA- vibro-compaction, (**e**) BS- Proctor compaction, (**f**) BS- vibro-compaction, (**g**) BFS- Proctor compaction, (**h**) BFS- vibro-compaction.

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
