# Peer review of "Geotechnical Properties of Anthropogenic Soils in Road Engineering"

_sustainability, doi:10.3390/su12124843_

Round 1
Reviewer 1 Report
I recommend the paper “Geotechnical Properties of Anthropogenic Soils in Road Engineering” only after major revision. It contains many inaccuracies that should be improved. The paper should be extensive revised and resubmitted for review.
The Introduction, material description and results should be reorganised. Too much space was devoted to the explanation of one of the waste in comparison to other, which can be shown in the reference list too. Reference items quoted in the paper are in a big mess. In many places Authors quoted items completely unconnected with the text in the paragraph. The test results are presented in a way that can cover their implementation inaccuracies. Detailed reservations will be given below in the following sections.
- In the Abstract, the Introduction and the Materials and Methods Authors stubbornly call CBR bearing capacity tests or even actual soil bearing capacity (line 54). In literature we can find many items describing the difference between them. CBR is used for evaluation the resistance to failure or indicate the load-carrying capacity. CBR does not reflect the shear stresses in pavement design; the shear stress depends on many factors but none of them is fully controlled or modelled in CBR test. The CBR abbreviation is not developed in any place.
- In the Introduction Authors stated that “The reclaimed concrete and slag materials as MM soils may differ significantly from NA. Therefore, in the soil classification, they are separated from NA and have been named as anthropogenic soils (AS) [11]”. And in all paper they try to prove a similarity of waste materials to natural soils. They call tested wastes as “blends”, which is wrong.
- In the Introduction Authors described three different tested wastes but literature items quoted in this place [12-20] concerns in one case of blast furnace slag, in 8 cases – recycled concrete aggregate and in none of them electric-power engineering wastes. The Introduction is a mix of general statement concerning the tested wastes properties and test methods. In my opinion it should be dedicated to the specified waste usability in the road pavement construction, in accordance with the title of the paper.
- In the Introduction and the Oedometric test results we can find many formulas not used in the paper, e.g. (2), (3), (5), (6), (7), (8). For example, Authors described formula for calculation of the compression index, Cc, for cohesive soil – all tested wastes are non-cohesive material. Some of formulas serve as indirect method for obtaining Cc, but Authors carried out laboratory tests. I cannot find the justifications for leaving them in the text.
- Authors called tested electric-power industry waste as “bottom slag”. Unfortunately, there is no such a waste. Judging by the appearance of waste in Figure 1c, we are dealing with “bottom ash” or “fly ash and bottom ash mix”. We should know from which place of the technological system this waste come.
- In the Materials and Methods Authors described recycled concrete aggregate, professionally. This remark does not concern to “bottom slag” and blast furnace slag. Their description is chaotic and contains factual errors. Authors states about production electrical from hard and brown coal, but do not mention about origin of tested waste(kind of coal). The values of specific gravity for “bottom slag” and blast furnace slag are disputable. The should be supported by literature values.
- The most important reservations concern the presentation of research results:
- Authors tested three different wastes and natural soil, but Figure 2 presents only one curve (green line), and additionally we do not which. Only for one soil we have calculated coefficients of gradation. All tested soils should be given grain-size distribution curves.
- Figure 3 presents the results of compaction test conducted by the Standard Proctor method and vibration method. I do not why we do not have compaction curves but only points. May be connection of these points will show that tests were not enough correct. All figures in Figure 3 should have shown the compaction curves with determination of maximum dry density and optimum water content. Compaction parameters are given for maximum of curves, but they are not the highest point in figure.
- Authors noticed in case of “bottom slag” that “during compaction water was leaking from between the cylinder slots” (sentence incorrect, additionally). “Therefore, during the field compaction, the BS should be in “flushed” conditions to obtain the highest degree of densification”. I have irresistible impression that it was the first compaction test for Authors.
- Results of static CBR test are given in Table 1 only for penetration 2.54 mm (?). They are shown in relation with moisture content. Please change presentation of the results and show the relationships in figures, because it will cause easier concluding and allow us asses the correctness of tests.
- In my opinion compression tests are not correct. “Each increment of loading was held constant for 100 seconds” (line 213), when they should be done to sample height But by the way presentation of test results should be done as compression curves for each soil. We have only one curve, and we do not know for which soil.
- Authors obtained for tested compacted soil preconsolidation pressure. Preconsolidation pressure is the maximum effective vertical overburden stress that a particular soil sample has sustained in the past. Please point a kind of overburden stress for soil without history of loading. According to Authors tested waste are overconsolidated (!).
- The Conclusion can be estimated only after improving all errors.
- Literature quotation errors:
- Literature item [35] was not quoted in the text.
- Item [36] quoted in line 95 is different item in Reference list: (36. Rowe, P.W., Barden, L. A new consolidation cell. J. Geotech. 1996, 16(2), 162-170). Proper date of Rowe and Barden is 1966.
- In line 129 Authors describe recycled concrete aggregate and quote items [36, 37]. Both of them do not concern to RCA, similarly as next item [39].
- In line 150 we have next error with quotation. Item [40] does not concern “bottom slag”, similarly as item [41] − blast furnace slag.
- In lines 294-296 in test result analysis we have text: “For example, the oedometer apparatus developed by Rowe and Barden is available for specimens with the diameters ranging from 75mm to 254mm (BSI 1990) [36, 37]”. Authors tested compression in CBR mould. We do not have this information in this place, but we have off-topic remark about size of unused hydraulic consolidometer.

Author Response
Respond for Review Report 1 for submission of a paper to Sustainability – MDPI journal
21.05.2020
Dear Reviewer,
We would like to thank You for all Your kind remarks. Please, find below the responses to Comments and Suggestions.
“The Introduction, material description and results should be reorganized. Too much space was devoted to the explanation of one of the waste in comparison to other, which can be shown in the reference list too. Reference items quoted in the paper are in a big mess. In many places Authors quoted items completely unconnected with the text in the paragraph. The test results are presented in a way that can cover their implementation inaccuracies. Detailed reservations will be given below in the following sections.”
We reorganized the material description and we presented it in the Paragraph 2. The literature citations were revised and now they are cited correctly as the reviewer suggested.
“In the Introduction Authors stated that “The reclaimed concrete and slag materials as MM soils may differ significantly from NA. Therefore, in the soil classification, they are separated from NA and have been named as anthropogenic soils (AS) [11]”. And in all paper they try to prove a similarity of waste materials to natural soils. They call tested wastes as “blends”, which is wrong.”
By pointing out this difference we wanted to show the particular difference between the anthropogenic soil and natural aggregate which is its origin. Therefore, based on this definition one can assume that the properties of the anthropogenic soil are different from the natural aggregates. We show that it is true but we can still unified this behavior in reference to the methods allocated in a classical geotechnics and in some cases, these differences are small. We added in the manuscript a proper explanation of this matter.
“In the Introduction Authors described three different tested wastes but literature items quoted in this place [12-20] concerns in one case of blast furnace slag, in 8 cases – recycled concrete aggregate and in none of them electric-power engineering wastes. The Introduction is a mix of general statement concerning the tested wastes properties and test methods. In my opinion it should be dedicated to the specified waste usability in the road pavement construction, in accordance with the title of the paper.”
The mentioned literature covered BFS (3 articles) BS (2 articles) and RCA (5 articles) and the reason for that was to point out that the CBR value as well as another properties were studied extensively. We have extended this paragraph and presented the CBR values from these studies to give some background to the results of our studies, as well as to fulfill the reviewer remark about dedication of literature review to the waste usability.
“In the Introduction and the Oedometric test results we can find many formulas not used in the paper, e.g. (2), (3), (5), (6), (7), (8). For example, Authors described formula for calculation of the compression index, Cc, for cohesive soil – all tested wastes are non-cohesive material. Some of formulas serve as indirect method for obtaining Cc, but Authors carried out laboratory tests. I cannot find the justifications for leaving them in the text.”
The equations (2) and (3) shows currently available formulas of resilient modulus calculation and we are referring into them when the equations (11) and (12) are derived. Therefore, we would like to leave them as the reference to our proposition. The Equation (8) is derived based on the test results and the Equations (5), (6) and (7). We indicated that some parameters may be also used for calculation of CC for the soil tested in this study, as the reviewer suggested.
“Authors called tested electric-power industry waste as “bottom slag”. Unfortunately, there is no such a waste. Judging by the appearance of waste in Figure 1c, we are dealing with “bottom ash” or “fly ash and bottom ash mix”. We should know from which place of the technological system this waste come.”
The name was corrected as the reviewer suggested. The origin of the soil was also added into text.
“In the Materials and Methods Authors described recycled concrete aggregate, professionally. This remark does not concern to “bottom slag” and blast furnace slag. Their description is chaotic and contains factual errors. Authors states about production electrical from hard and brown coal, but do not mention about origin of tested waste(kind of coal). The values of specific gravity for “bottom slag” and blast furnace slag are disputable. The should be supported by literature values.”
The description of the remarked issues was applied in the manuscript. The specific gravity value for each soil was estimated by the laboratory tests with use of the piknometer method.
“Authors tested three different wastes and natural soil, but Figure 2 presents only one curve (green line), and additionally we do not which. Only for one soil we have calculated coefficients of gradation. All tested soils should be given grain-size distribution curves.
In this article the tested soil material was first sieved into fractions and then the universal soil grain composition was created for each kind of soil. The aim of such procedure was to compare the mechanical properties of soils with the same grain composition. This remark was added to the manuscript.
“Figure 3 presents the results of compaction test conducted by the Standard Proctor method and vibration method. I do not why we do not have compaction curves but only points. May be connection of these points will show that tests were not enough correct. All figures in Figure 3 should have shown the compaction curves with determination of maximum dry density and optimum water content. Compaction parameters are given for maximum of curves, but they are not the highest point in figure.”
Thank you for this remark. We added the compaction curves. The non-cohesive soils have lower susceptibility to the water content than the cohesive soils. This property can be observed as an absence of typical compaction curve characteristic. The Proctor method was proven to be a wrong method for OMC estimation due to high inconsistence of the test results. Therefore, the inconsistence is the result of the soil properties and not of the error in the tests procedure. Consequently, we were forced to use compaction curves to estimate the soil OMC. We could have removed some points from the plot due to the fact that some might go wrong during the test but such solution would be misleading, in our opinion. Nevertheless, the reviewer rightly pointed out that the absence of compaction curves is not a good solution to these deliberations. We added the compaction curves as reviewer suggested, as well as an extended comment in the manuscript.
“Authors noticed in case of “bottom slag” that “during compaction water was leaking from between the cylinder slots” (sentence incorrect, additionally). “Therefore, during the field compaction, the BS should be in “flushed” conditions to obtain the highest degree of densification”. I have irresistible impression that it was the first compaction test for Authors.”
The mentioned sentence was removed from the manuscript.
“Results of static CBR test are given in Table 1 only for penetration 2.54 mm (?). They are shown in relation with moisture content. Please change presentation of the results and show the relationships in figures, because it will cause easier concluding and allow us asses the correctness of tests.
The CBR test results are changed, according to the reviewer comment.
“In my opinion compression tests are not correct. “Each increment of loading was held constant for 100 seconds” (line 213), when they should be done to sample height But by the way presentation of test results should be done as compression curves for each soil. We have only one curve, and we do not know for which soil.”
Thank you for this remark. We decided to present 56 curves in as a 3D graphic form in Annex A and we are presenting them here as well. 100 seconds is enough time to reach the final settlements of the soil under certain load and to avoid the time-effect, which is connected to grain crushing during loading.
Figure Compression curves od soils in different moisture content (a) NA – Proctor compaction, (b) NA- vibro-compaction, (c) RCA- Proctor compaction, (d) RCA- vibro-compaction, (e) BS- Proctor compaction, (f) BS- vibro-compaction, (g) BFS- Proctor compaction, (h) BFS- vibro-compaction.
“Authors obtained for tested compacted soil preconsolidation pressure. Preconsolidation pressure is the maximum effective vertical overburden stress that a particular soil sample has sustained in the past. Please point a kind of overburden stress for soil without history of loading. According to Authors tested waste are overconsolidated (!).”
As a matter of fact, the preconsolidation origin may be not only natural but as well anthropogenic. There is a lot of scientific reports which shows overconsolidation pressure from traffic or human activities. In this study, preconsolidation is the result of the compaction.
“Literature quotation errors”
We checked the literature and we have updated it as reviewer suggested.
We checked the manuscript for typos, and complete changes are presented in the manuscript. We present all changes in the red font style.
Thank You again for Your valuable remarks on our manuscript.
Sincerely,
Andrzej, Kasia, Emil, Raimondas and Wojciech

Reviewer 2 Report
The paper is well orientated, with a clear and logical application meaning.
The proposed methodology is well planned and adapted to the application field, with clear concept and interesting ideas to advance in a more science based concept. (Not easy in this field work).
The proposed alternative materials are consistent with the application, anyway I think that it is not clear the viability of the proposal. Some numbers of the volume of Bottom Slags are presented, but it would be interesting a improved information on the relative volume that each proposed alternative material can represent in the whole used material used to this objective. It is difficult to understand if the proposal represent an change in the market or only a anecdotical case.
The second point that can improve the paper is to increase the definition of the used raw materials. There are lots of options of RCA, or slags, and only a minor bibliography information is done. I thinks that a more detailed, characterization over than density can improve the information, absorption, fines, adhered mortar... or details in the composition or quality or other parameters defining the aggregates quality for the proposed use.
There is not one type of RCA, or slag, and with your paper may be is not good to avoid a risk of generalization.
Author Response
Respond for Review Report 2 for submission of a paper to Sustainability – MDPI journal
21.05.2020
Dear Reviewer,
We would like to thank You for all Your kind remarks. Please find below the responses to Comments and Suggestions.
“The proposed alternative materials are consistent with the application, anyway I think that it is not clear the viability of the proposal. Some numbers of the volume of Bottom Slags are presented, but it would be interesting a improved information on the relative volume that each proposed alternative material can represent in the whole used material used to this objective. It is difficult to understand if the proposal represent an change in the market or only a anecdotical case.”
We have tested four different types of soil material with the same gradation characteristics to evaluate the differences between natural aggregate and antropogenic soils, based on the common tests used in the road enginnering field. Therefore, our proposal, was aimed to show these differences and as reviewer pointed out to show that alternative material may be used as a replacement of natural aggregates. We added in the manuscript a proper explanation in the introduction paragraph.
“The second point that can improve the paper is to increase the definition of the used raw materials. There are lots of options of RCA, or slags, and only a minor bibliography information is done. I thinks that a more detailed, characterization over than density can improve the information, absorption, fines, adhered mortar... or details in the composition or quality or other parameters defining the aggregates quality for the proposed use. ”
To fulfill the reviewer's suggestion we extended the description of the materials in Paragraph 2.
“There is not one type of RCA, or slag, and with your paper may be is not good to avoid a risk of generalization.”
Thank you for this remark. We pointed this out in the Conclusion paragraph.
We checked the manuscript for typos, and complete changes are presented in the manuscript. We present all changes in the red font style.
Thank You once again for Your valuable remarks on our manuscript.
Sincerely,
Andrzej, Kasia, Emil, Raimondas and Wojciech
Round 2
Reviewer 1 Report
I recommend the paper “Geotechnical Properties of Anthropogenic Soils in Road Engineering” still only after major revision. The Authors improved paper in the case of many problems, but in some cases, they omitted them or gave wrong explanation. The paper should be revised and resubmitted for review.
The presentation of research results still needs improvement. My improvement requests in the first step of reviewing were ignored in points: 7a, 7e-f, 9e.
In point 7a I noticed that Figure 2 was concerned only for one type of tested waste. All wastes should be described by their graining. The „solid green line” was given for only one anthropogenic soil. All four tested soils should be given grain-size distribution curves and calculated coefficients of gradation.
In point 7e I stated that the presented compression tests are not correct. “Each increment of loading was held constant for 100 seconds” (now line 260), when they should be done to sample height stabilization. Authors explained that 100 seconds is enough time to reach the final settlements of the soil under certain load and to avoid the time-effect, which is connected to grain crushing during loading. In my opinion crushing took place mainly under too great loading, but not in case of long time (fatigue crushing needs greater periods of times). Please show a figure with consolidation curves to prove enough settlement. In the same point I paid attention for wrong presentation of test results because it should have been done as compression curves for each soil. We still have only one curve, but sample description was supplemented.
Concerned point 7f. The Authors insist that preconsolidation pressure can be conducted for newly compacted soil. The Authors explained: „There is a lot of scientific reports which shows overconsolidation pressure from traffic or human activities. In this study, preconsolidation is the result of the compaction”. This explanation does not concern the described problem. Preconsolidation pressure is the maximum effective vertical overburden stress that a soil sample has sustained in the past. According my knowledge and experience preconsolidation of newly compacted soils is impossible, especially for soil compacted by vibration as in the paper. The phenomenon of overconsolidation of compacted soil can only take place for testing samples after static compaction. Please prove your point by presenting literature examples from known books or journal papers (with high impact factor).
In point 9e I pointed that in test result analysis the Authors had written (now in lines 348-349): “For example, the oedometer apparatus developed by Rowe and Barden is available for specimens with the diameters ranging from 75mm to 254mm (BSI 1990) …”. Authors tested compression in CBR mould. We do not have this information in this place, but we have off-topic remark about size of unused hydraulic consolidometer. The Authors ignored my earlier remark and stated that they „checked the literature and we have updated it as reviewer suggested”.
The Conclusion can be estimated only after improving all errors.

Author Response
Respond for Review Report 1 for submission of a paper to Sustainability – MDPI journal
04.06.2020
Dear Reviewer,
We would like to thank You for all Your kind remarks. Please, find below the responses to Comments and Suggestions.
“In point 7a I noticed that Figure 2 was concerned only for one type of tested waste. All wastes should be described by their graining. The „solid green line” was given for only one anthropogenic soil. All four tested soils should be given grain-size distribution curves and calculated coefficients of gradation..”
In this study, we conducted the tests for 4 types of soil, and each soil had precisely the same grain composition, such a procedure aimed to be able to compare the properties of the tested soil directly. Therefore, the solid green line represents all 4 types of tests soils. We might say that the soil gradation was universal for all 4 soil types.
“In point 7e I stated that the presented compression tests are not correct. “Each increment of loading was held constant for 100 seconds” (now line 260), when they should be done to sample height stabilization. Authors explained that 100 seconds is enough time to reach the final settlements of the soil under certain load and to avoid the time-effect, which is connected to grain crushing during loading. In my opinion crushing took place mainly under too great loading, but not in case of long time (fatigue crushing needs greater periods of times). Please show a figure with consolidation curves to prove enough settlement. In the same point I paid attention for wrong presentation of test results because it should have been done as compression curves for each soil. We still have only one curve, but sample description was supplemented.”
The presented 100-second rule was chosen because, during that time, the primary compression which is caused by the new loading step is over. On Figure 1 we present the raw data from the tested sample. The careful analysis shows that in all cases, the sample height stabilization occurs during the 100 seconds. What is more, the secondary compression takes place, which can be described by a linear relationship between the displacement and log of time. It indicates that the time effect is connected to the grain crushing. To avoid misleading, we appended additional explanations in the text concerning the discussed matter. In a previous response to Reviewer, we presented compression curves in Appendix A (now Apendix B). In this document, we supplement all compression curves in .pdf file
“Concerned point 7f. The Authors insist that preconsolidation pressure can be conducted for newly compacted soil. The Authors explained: „There is a lot of scientific reports which shows overconsolidation pressure from traffic or human activities. In this study, preconsolidation is the result of the compaction”. This explanation does not concern the described problem. Preconsolidation pressure is the maximum effective vertical overburden stress that a soil sample has sustained in the past. According my knowledge and experience preconsolidation of newly compacted soils is impossible, especially for soil compacted by vibration as in the paper. The phenomenon of overconsolidation of compacted soil can only take place for testing samples after static compaction. Please prove your point by presenting literature examples from known books or journal papers (with high impact factor).”
Thank you for this remark. The performed oedometric tests have shown that the preconsolidation pressure can be calculated. In our opinion, the tested soil behaves a phenomenon that can be recognized as preconsolidation if we would put aside the preconsolidation definition. Therefore, our aim is to report this finding despite the well-known definition. What is more, we believe that the preconsolidation pressure which existence was reported here is caused not by the maximum effective vertical overburden stress but by the energy of compaction. The effective stress is as well the energy imposed on the sample, and we believe that the energy of compaction may have the same effect as static stress imposed on the sample. The preconsolidation is a certain skeleton arrangement able to sustain the energy of maximal loading, is different from the Vibro-compaction actions in terms of load characteristics. Nevertheless, the soil skeleton during the vibro-compaction also was changed to sustain such vibration energy. The exact mechanics of Vibro compaction was presented in the Basic of Foundation Design by Fallenius and in the article of Duan et al. (2019). According to the literature, the preconsolitation effect is caused by high horizontal stress in the soil, which are the effect of strong horizontal pulses. Therefore, we believe that the problem is in the name of the observed phenomena, which the Reviewer was right to point out. To resolve this issue, we decided to change the preconsolidation term to “apparent preconsolidation” term. An appropriate explanation was appended in the manuscript.
In point 9e I pointed that in test result analysis the Authors had written (now in lines 348-349): “For example, the oedometer apparatus developed by Rowe and Barden is available for specimens with the diameters ranging from 75mm to 254mm (BSI 1990) …”. Authors tested compression in CBR mould. We do not have this information in this place, but we have off-topic remark about size of unused hydraulic consolidometer. The Authors ignored my earlier remark and stated that they „checked the literature and we have updated it as reviewer suggested”.
The information about the cylinder type was appended in point 2.4. To fulfil the Reviewer's suggestion, we appended the diameter and height of the cylinder at this point, as well as mentioned by the reviewer part of the manuscript.
Thank you for your remarks on this manuscript.
Sincerely,
Andrzej Kasia, Emil, Raimondas and Wojciech

Round 3
Reviewer 1 Report
I accept the paper in the present form Authors improved it by including additional explanation and test results.